# CONTINUAL DENSITY RATIO ESTIMATION (CDRE): A NEW METHOD FOR EVALUATING GENERATIVE MODELS IN CONTINUAL LEARNING

## ABSTRACT

We propose a new method Continual Density Ratio Estimation (CDRE), which can estimate density ratios between a target distribution of real samples and a distribution of samples generated by a model while the model is changing over time and the data of the target distribution is not available after a certain time point. This method perfectly fits the setting of *continual learning*, in which one model is supposed to learn different tasks sequentially and the most crucial restriction is that model has none or very limited access to the data of all learned tasks. Through CDRE, we can evaluate generative models in continual learning using $f$-divergences. To the best of our knowledge, there is no existing method that can evaluate generative models under the setting of continual learning without storing real samples from the target distribution.

## 1 INTRODUCTION

Density Ratio Estimation (DRE) (Sugiyama et al., 2012) is a methodology for estimating the density ratio between two probability distributions and it can be applied to two-sample tests in which only samples of the two distributions are available. It has a wide range of applications in machine learning, such as distribution comparison, mutual information estimation, outliers detection, etc.. However, under certain restrictive conditions in the real world, i.e. one distribution is changing over time and the samples of the other distribution are not available after some time point, the existing methods of DRE are not applicable any more which leads to the unavailability of those applications of DRE as well. In order to enable DRE under such restrictive conditions, we propose a novel method Continual Density Ratio Estimation (CDRE) which can estimate the density ratio between a target distribution and a model distribution without storing any samples from target distribution while the model distribution is changing over time.

There can be diverse reasons in the real world of limiting access to the raw data after a model is trained on it. For example, researchers of a hospital may have trained a model for one type of disease and the raw data of patients cannot be shared with other institutions, if they want to collaborate with another institution to enable the model capable of a similar type of disease as well, the model can only be incrementally trained on new data without sharing the previous data. Besides the privacy issue, limited cost budget can be another cause of such a problem, such as the data storage cost is quite high, or the data is not available for free after its copyright has expired.

The restrictive conditions described above perfectly match the problem setting of *continual learning* in which a single model evolves over time by learning new tasks sequentially with none or very limited access to the data of old tasks and yet is able to perform on all seen tasks. The most crucial obstacle of continual learning is that a model tends to forget previous tasks while learning a new task, which is a phenomenon called *catastrophic forgetting* (Kirkpatrick et al., 2017). Generative models play an important role in continual learning because they can be employed to help other models with keeping memories by generating samples of previous tasks (Shin et al., 2017; Wu et al., 2018), a method known as "generative replay". A simplified scenario for generative models in continual learning is depicted in Fig. 1, where the goal is to learn a generative model for one category (digit) per task, but still be able to generate samples of all previous categories. The training dataset of task $i$

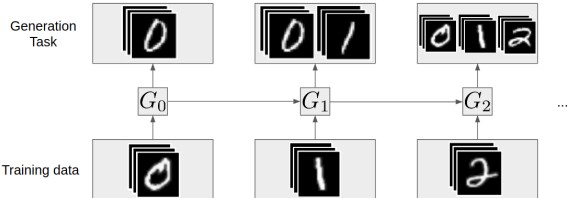

Figure 1: Demonstration of generative models in continual learning. At task $i$ the training set consists of samples of category $i$ and samples generated by the model at the previous task, and the task is to generate samples from all previously seen categories (figure reproduced from Lesort et al. (2018)).

consists of real samples of category $i$ and samples of task $1 \cdots i-1$ generated by the previous model.

Despite the importance of generative models in continual learning, there is no effective way to evaluate them under the restrictions of continual learning. The related works (Wu et al., 2018; Shin et al., 2017; Lesort et al., 2018) either evaluate them by classification performance, or just displaying some model samples to let readers judge them visually, or applying usual measures of generative models in static learning by comparing with real samples from the target distribution. However, the performance of generative models is not always tied to the performance of a classification task, it may be decided by the fidelity of model samples as in many applications of generative models (Brock et al., 2019; Karras et al., 2019; van den Oord et al., 2017). Moreover, it is questionable how many real samples we can obtain to evaluate generative models in continual learning. A small sample size (less than a few hundreds) as commonly used in episodic memories (Lopez-Paz et al., 2017; Chaudhry et al., 2019) or coresets (Nguyen et al., 2018) cannot guarantee the accuracy of measures for evaluating generative models.

The principal idea of evaluating generative models is estimating the difference between the target distribution and the model distribution, for example, Fréchet Inception Distance (FID) (Heusel et al., 2017) and Kernel Inception Distance (KID) (Bińkowski et al., 2018) estimate Wasserstein-2 distance and Maximum Mean Discrepancy (MMD), respectively. Likewise, $f$-divergences is a well-studied family of divergences that are commonly used to measure differences between two distributions, more importantly, one can estimate $f$-divergences by estimating density ratios.

We show that our new method CDRE can effectively estimate $f$-divergences under the setting of continual learning. Consequently, we can evaluate generative models in continual learning using $f$-divergences estimated by CDRE, which is supported by experiment results in comparison with commonly used FID, KID. In the absence of prior work, we also provide empirical analysis of differences between FID, KID and $f$-divergences in terms of evaluating generative models.

## 2 PRELIMINARIES

In this section we introduce the basic formulation of Density Ratio Estimation (DRE) as it is the foundation of CDRE, and we discuss estimating $f$-divergences by DRE as well.

### 2.1 DENSITY RATIO ESTIMATION (DRE).

There are two principal methods for DRE introduced in Sugiyama et al. (2012): Kullback-Leibler Importance Estimation Procedure (KLIEP) and Least-Squares Importance Fitting (LSIF), which are based on Kullback-Leibler (KL) divergence and Pearson ($\chi^2$) divergence, respectively. Here we review the formulation of KLIEP, which will form a building block of our method. Let $r^*(x) = \frac{p(x)}{q(x)}$ be the (unknown) true density ratio, then $p(x)$ can be estimated by $\tilde{p}(x) = r(x)q(x)$, where $r^*(x)$ is modeled by $r(x)$. Hence, we can optimize $r(x)$ by minimizing the KL-divergence between $p(x)$ and $\tilde{p}(x)$ with respect to $r$:

$$D_{KL}\left(p(x)||\tilde{p}(x)\right) = \int p(x)\log\frac{p(x)}{\tilde{p}(x)}dx = \int p(x)\log r^*(x)dx - \int p(x)\log r(x)dx \quad (1)$$

where $r(x)$ satisfies $r(x) > 0$ and $\int r(x)q(x)dx = \int \tilde{p}(x)dx = 1$. The first term of the right side in Eq. (1) is a constant w.r.t. $r(x)$, then the objective of optimizing $r(x)$ is as below:

$$J_r = \max_r \frac{1}{N} \sum_{i=1}^{N} \log r(x_i), \ \ x_i \sim p(x), \ \ s.t. \ \ \frac{1}{M} \sum_{j=1}^{M} r(x_j) = 1, \ \ x_j \sim q(x), \ \ \text{and} \ \ r(x) \geq 0, \ \ \forall x. \quad (2)$$

One convenient way of parameterizing $r(x)$ is by using a log-linear model with normalization, which then automatically satisfies the constraints in Eq. (2):

$$r(x;\beta) = \frac{\exp(\psi_\beta(x))}{\frac{1}{M} \sum_{j=1}^{M} \exp(\psi_\beta(x_j))}, \quad x_j \sim q(x), \quad \psi_\beta : \mathbb{R}^D \rightarrow \mathbb{R}, \quad (3)$$

where $\psi_\beta$ can be any deterministic function and we use a neural network as $\psi_\beta$ in our implementations, $\beta$ representing parameters of the neural network.

## 2.2 ESTIMATING $f$-DIVERGENCES BY DENSITY RATIO ESTIMATION

The definition of $f$-divergences is given in Eq. (4), where $f$ is a convex function and satisfies $f(1) = 0$. Given a specified $f$, we can estimate the empirical divergence using density ratios without knowing the density functions. For example, taking $f(r) = r \log r$ recovers the KL divergence.

$$\mathscr{D}_f(p(x)||q(x)) = \mathbb{E}_{q(x)}\left[f\left(\frac{p(x)}{q(x)}\right)\right] \approx \frac{1}{N} \sum_{i=1}^{N} f(r(x_i)), \quad x_i \sim q(x), \quad r(x_i) = \frac{p(x_i)}{q(x_i)} \quad (4)$$

Estimating $f$-divergences by density ratios has been well studied (Kanamori et al., 2011; Nguyen et al., 2010). Although it can also be estimated by the density functions of distributions, that is beyond the scope of this paper and density functions are often not available.

One concern of estimating $f$-divergences by DRE is from high dimensional data which might require large sample size to achieve convergence (Rubenstein et al., 2019). One way to mitigate the problem is to perform dimensionality reduction in combination with DRE, for which several methods have been introduced in Sugiyama et al. (2012). A fundamental assumption of these methods is that the difference between two distributions can be confined in a subspace, which means $p(z)/q(z) = p(x)/q(x)$ where $z$ is a lower-dimensional representation of $x$. This is aiming for exact density ratio estimation, in another word, if the difference is large in the original space, it is still large in the subspace. Besides the expensive cost of searching such a subspace, significant differences between distributions still cause unrealistic convergence rate of DRE methods. For example, the convergence analysis of KLIEP is described in Sugiyama et al. (2008), as in short, the convergence rate depends on the bounds of estimated ratios, as well as the bounds of the entropy of estimated ratios. Hence, it is desirable to scale down the difference between high dimensional distributions and make ratios well bounded for a easier convergence by dimensionality reduction, whilst ensuring that the estimated divergence is still a faithful reflection of the true divergence. On the other hand, DRE can work with high-dimensional data when the two distributions are close to each other. We demonstrate this by experiments with high-fidelity model samples in a high-dimensional space (Appx. B.2).

According to the information monotonicity of $f$-divergences (Amari, 2009), we can estimate a lower bound of the $f$-divergence on an arbitrary surrogate feature space: $D_f(q(z)||p(z)) \leq D_f(q(x)||p(x))$, where $p(z) = \int p(x)p(z|x)dx, q(z) = \int q(x)p(z|x)dx$ and $p(z|x)$ is an arbitrary transition probability. Using a surrogate feature space is a widely applied technique in measurements of generative models. For instance, the inception feature defined for Inception Score (IS) (Salimans et al., 2016) can be viewed as from a surrogate feature space and it is widely applied in other measures of generative models (such as FID, KID, Precision and Recall for Distributions (PRD)) as well. Analogously, we introduce Continual Variational Auto Encoder (CVAE) as a solution of generating lower dimensional features for CDRE in the later section.

## 3 CONTINUAL DENSITY RATIO ESTIMATION

In this section We first introduce the general formulation of CDRE by only considering one target distribution through out the whole time. After that, we introduce the formulation under the full setting of continual learning in which case the model learns multiple target distributions in a sequential manner.

Let $X$ denote the data from the target distribution and $p(x)$ denote its density function. Note that $p(x)$ will not change over time but $X$ will be unavailable when $t > 1$, where $t$ represents the index of current time steps. $G_t$ denotes a generative model at time $t$, $\hat{X}_t$ denotes samples generated by $G_t$ and $q_t(x)$ denotes the density function of $\hat{X}_t$, which are all changing while $t$ changing. The goal of CDRE is to estimate the density ratio $p(x)/q_t(x), \forall t$, which can be decomposed as follows:

$$r_t(x) = \frac{p(x)}{q_t(x)} = \frac{q_{t-1}(x)}{q_t(x)} \frac{p(x)}{q_{t-1}(x)} = r_{s_t}(x) r_{t-1}(x), \tag{5}$$

where $r_{s_t}(x) = q_{t-1}(x)/q_t(x)$ represents the empirical density ratio of model samples obtained at two adjacent time steps. This decomposition gives a method to estimate $p(x)/q_t(x)$ iteratively, without the needs of keeping raw data samples from $p(x)$ as $t$ increases. The key point is that we can optimize $r_t(x)$ by optimizing $r_{s_t}(x)$ when $r_{t-1}(x)$ is known. Existing methods for DRE, such as KLIEP introduced in Sec. 2, can be applied to estimating $r_1(x)$ and $r_{s_t}(x)$ as the basic ratio estimator of CDRE. Furthermore, only one extra constraint is required:

$$\int r_{s_t}(x) q_t(x) dx = \int \frac{r_t(x)}{r_{t-1}(x)} q_t(x) dx = 1 \tag{6}$$

For instance, when $r_t(x)$ is defined by using the log-linear form as in Eq. (3), $r_{s_t}$ can be expressed as follows, where $\beta_t$ represents parameters of $r_t(x)$:

$$r_{s_t} = \frac{r_t(x)}{r_{t-1}(x)} = \exp\{\psi_{\beta_t}(x) - \psi_{\beta_{t-1}}(x)\} \times \frac{\frac{1}{N_{t-1}} \sum_{j=1}^{N_{t-1}} \exp\{\psi_{\beta_{t-1}}(x_{t-1,j})\}}{\frac{1}{N_t} \sum_{i=1}^{N_t} \exp\{\psi_{\beta_t}(x_{t,i})\}}, \tag{7}$$

$$x_{t,i} \sim q_t(x), \quad x_{t-1,j} \sim q_{t-1}(x)$$

When $r_{s_t}$ satisfies the constraint in Eq. (6), we have following equality by replacing Eq. (7) into Eq. (6) in which we approximate the expectation using Monte Carlo integration:

$$\frac{\frac{1}{N_t} \sum_{i=1}^{N_t} \exp\{\psi_{\beta_t}(x_{t,i})\}}{\frac{1}{N_{t-1}} \sum_{j=1}^{N_{t-1}} \exp\{\psi_{\beta_{t-1}}(x_{t-1,j})\}} = \frac{1}{N_t} \sum_{i=1}^{N_t} \exp\{\psi_{\beta_t}(x_{t,i}) - \psi_{\beta_{t-1}}(x_{t,i})\}, \tag{8}$$

$r_{s_t}$ can then be rewritten as:

$$r_{s_t} = \frac{\exp\{\psi_{\beta_t}(x) - \psi_{\beta_{t-1}}(x)\}}{\frac{1}{N_t} \sum_{i=1}^{N_t} \exp\{\psi_{\beta_t}(x_{t,i}) - \psi_{\beta_{t-1}}(x_{t,i})\}}, \tag{9}$$

In this manner, $r_{s_t}$ is in the same log-linear form of Eq. (3). We can directly apply KLIEP to optimize $\beta_t$, which gives the following objective:

$$\max_{\beta_t} \mathscr{L}_t(x; \beta_t) = \max_{\beta_t} \frac{1}{N_{t-1}} \sum_{j=1}^{N_{t-1}} \log r_{s_t}(x_{t-1,j}) + \lambda_c (\Psi_t^{t-1}(X_t) \times \Psi_{t-1}(X_{t-1}) / \Psi_t(X_t) - 1)^2,$$

$$\text{where} \quad \Psi_t(X_t) = \frac{1}{N_t} \sum_{i=1}^{N_t} \exp\{\psi_{\beta_t}(x_{t,i})\}, \quad \Psi_t^{t-1}(X_t) = \frac{1}{N_t} \sum_{i=1}^{N_t} \exp\{\psi_{\beta_t}(x_{t,i}) - \psi_{\beta_{t-1}}(x_{t,i})\} \tag{10}$$

where $\beta_{t-1}$ can be viewed as constant since it has been learned at task $t-1$. The equality constraint of Eq. (8) has been transformed and put into the objective using a hyperparameter $\lambda_c$. We observe there is a trade-off between bias and variance controlled by $\lambda_c$. Larger value of $\lambda_c$ results in smaller bias but larger variance. Relevant experimental results are demonstrated in Fig. 2. The density ratio estimator of KLIEP is an unbiased estimator when the constraint is satisfied. However, as we replace the hard constraint by a soft constraint, the larger $\lambda_c$ makes the estimator with soft constraint closer to the unbiased one, which leads to smaller bias. The bias is getting less when increasing the lambda, and as a trade off, the variance starts to increase.

It has been discussed in Mohamed & Lakshminarayanan (2016) that the discriminator of some type of Generative Adversarial Networks (GANs) can be viewed as a density ratio estimator. We have also applied the formulation of discriminators of $f$-GAN (Nowozin et al., 2016) to the basic ratio estimator in CDRE. Nonetheless, we found it is less robust than KLIEP and may not satisfy the constraint of density ratios as it is not defined for the purpose of estimating ratios. For example, the

ratio can be negative by the formulation of $\chi^2$-divergence in $f$-GAN. Therefore, we stick to KLIEP as the basic ratio estimator of CDRE in our experiments. However, if estimating some divergences which are not based on log-ratios it may be better try some other form of ratio estimators. For instance, it may be preferable to use LSIF when estimating Pearson $\chi^2$ divergence, since a small deviation in log-ratio can result in large differences. Also, since LSIF itself is based on Pearson $\chi^2$ divergence, it appears to be a more natural choice.

**CDRE in continual learning.** Now we consider the full setting in continual learning, in which the model needs to learn a new distribution at each time step $t$ and we refer to it as task $t$. Let $X_\tau$ denote the raw data from task $\tau$, which is not available at task $t$ when $t > \tau$, and let $p(x|\tau)$ denote the density function of $X_\tau$. Similarly, let $\hat{X}_{\tau,t}$ denote samples of task $\tau$ generated by the model at time $t$ ($G_t$) and $q_t(x|\tau)$ denote the density function of $\hat{X}_{\tau,t}$. Now we have

$$r_t(x|\tau) = \frac{p(x|\tau)}{q_t(x|\tau)} = \frac{q_{t-1}(x|\tau)}{q_t(x|\tau)}\frac{p(x|\tau)}{q_{t-1}(x|\tau)} = r_{s_t}(x|\tau)r_{t-1}(x|\tau), \quad r_{s_t}(x|\tau) = \frac{q_{t-1}(x|\tau)}{q_t(x|\tau)} \tag{11}$$

We optimize the estimator at time $t$ by an average objective

$$\max_{\beta_t} \mathscr{L}_t(\beta_t) = \max_{\beta_t} \frac{1}{t}\sum_{\tau=1}^{t} \mathscr{L}_t(x|\tau;\beta_t) \tag{12}$$

where $\mathscr{L}_t(x|\tau;\beta_t)$ is the same as $\mathscr{L}_t(x;\beta_t)$ in Eq. (10) for a given $\tau$, and $r_{s_t}(x|\tau)$ is defined by the same form of Eq. (9) except $\psi_{\beta_t}(x)$ has been replaced by $\psi_{\beta_t}(x,\tau)$. By concatenating the task index to each data sample as the input of a ratio estimator, we can avoid learning $T$ separate ratio estimators for $T$ tasks. In addition, we set the output of $\psi_{\beta_t}(\cdot)$ as a $t$-dimensional vector $\{o_1,\ldots,o_t\}$ where $o_\tau$ corresponds to the output of $\psi_{\beta_t}(x,\tau)$. We found that this improves the accuracy of the ratio estimation when $t$ increases because the model capacity increases as well.

**Estimating $f$-divergences by CDRE** We evaluate a generative model in continual learning by estimating the averaged $f$-divergences over all learned tasks:

$$\bar{\mathscr{D}}_t = \frac{1}{t}\sum_{\tau=1}^{t}\mathscr{D}_f(p(x|\tau)||q_t(x|\tau)) \tag{13}$$

It is noteworthy that estimation error obtained at each task will accumulate in CDRE as shown below, where $r_{s_t}^*$ denotes the true ratio and $\Delta r_{s_t}$ denotes the estimation error:

$$r_t(x|\tau) = r_{s_t}(x|\tau)r_{t-1}(x|\tau) = r_\tau(x|\tau)\prod_{i=\tau+1}^{t} r_{s_i}(x|\tau) = \prod_{i=\tau}^{t} r_{s_i}^*(x|\tau)\Delta r_{s_i}(x|\tau),$$
$$\text{where} \quad \tau \le t, \quad r_{s_i}(x|\tau) = r_{s_i}^*(x|\tau)\Delta r_{s_i}(x|\tau), \quad r_{s_\tau}(x|\tau) = r_\tau(x|\tau). \tag{14}$$

We observe that larger errors often lead to larger variance of the estimated $f$-divergences and larger errors are often caused by greater differences between two distributions. We give more analysis about the variance of the estimated $f$-divergences in Sec. 4 along with the experiment results.

**Feature generation for CDRE** As discussed in Sec. 2.2, we perform dimensionality reduction as a preprocessing of CDRE for extracting features from high dimensional data. A pre-trained classifier is often used to generate surrogate features for image data (e.g. inception features (Salimans et al., 2016)). However, we consider using Variational Auto Encoder (VAE) without pre-training in our experiments for two reasons: (a) it may be difficult to obtain an homogeneous dataset for pre-training; (b) it may not be able to train a classifier when there are no labels available. In order to cope with the setting of continual learning, we introduce Continual Variational Auto Encoder (CVAE) for feature generation in a pipeline with CDRE. The loss function of CVAE is accordingly adjusted by the principle of Variational Continual Learning (VCL) (Nguyen et al., 2018):

$$\mathscr{L}_t(\theta_t,\vartheta_t) = NLL + D_{KL}(q_t(z)||q_{t-1}(z)), \quad q_t(z) = \mathscr{N}(\mu_{\theta_t}(x),\sigma_{\theta_t}(x)),$$
$$NLL = -\frac{1}{t}\left[\sum_{\tau=1}^{t-1}\mathbb{E}_{q_{t-1}(x|\tau)}[\mathbb{E}_{q_t(z)}[\log p(x|z;\vartheta_t)]] + \mathbb{E}_{p(x|t)}[\mathbb{E}_{q_t(z)}[\log p(x|z;\vartheta_t)]]\right] \tag{15}$$

where $\theta_t$ and $\vartheta_t$ denote parameters of the encoder and decoder at task $t$, respectively. It is different with VAEs of continual learning described in Nguyen et al. (2018) because the posteriors of adjacent tasks in the KL divergence is w.r.t. latent codes $z$ rather than parameters $\theta$ of the VAE. In Nguyen et al. (2018) the encoder is task-specific which is computational costly and when the encoder is shared across tasks we found that latent codes cannot preserve much differences between different tasks. In our case, we just expect the encoder gives similar $z$ for a similar $x$ at different time $t$, which can guarantee the consistency of inputs for the previous estimator (i.e. $\psi_{\beta_{t-1}}(x_{t,i})$ in Eq. (10)). This adjusted objective can achieve the goal with cheaper cost in our experiments. We are aiming to provide a general solution of feature generation in practical situations, nevertheless, other commonly used methods (i.e. pre-trained classifiers) are also applicable to CDRE.

## 4 EXPERIMENTS

In the absence of prior work on evaluating generative models by $f$-divergences, we provide experimental results in Appx. B on comparing $f$-divergences with FID (Heusel et al., 2017), KID (Bińkowski et al., 2018) and PRD (Sajjadi et al., 2018) in a few toy experiments and a high-dimensional dataset of the real world. Through these experiments, we show that $f$-divergences can be alternative measures of generative models and one may obtain richer criteria by $f$-divergences.

In this section, we first show that CDRE can effectively estimate $f$-divergences in the setting of continual learning by experiments on synthetic data. We then evaluate WGAN (Arjovsky et al., 2017), WGAN-GP (Gulrajani et al., 2017) and several members of $f$-GAN (Nowozin et al., 2016) on two bench-mark datasets in continual learning: Fashion-MNIST (Xiao et al., 2017) and MNIST (LeCun et al., 2010). All GANs tested in continual learning are conditional GANs (Mirza & Osindero, 2014) with task indices as conditioners, and one task includes a single class of the dataset.

We have deployed two feature generators for experiments with MNIST and Fashion-MNIST: 1) A classifier which is a Convolutional Neural Network (CNN) trained on real samples of all classes, the extracted features are the activations of the last hidden layer (a similar setting is suggested in Bińkowski et al. (2018) for testing KID on MNIST); 2) A CVAE trained incrementally in the procedure of continual learning, and the features are the output of the encoder. The dimension of features are 64 for both classifier and CVAE. More details of the experimental settings are described in Appx. A. We use the classifier as the feature generator for FID,KID and PRD and use the CVAE as the feature generator for CDRE in all experiments except specified explicitly.

### 4.1 EXPERIMENTS WITH SYNTHETIC DATA

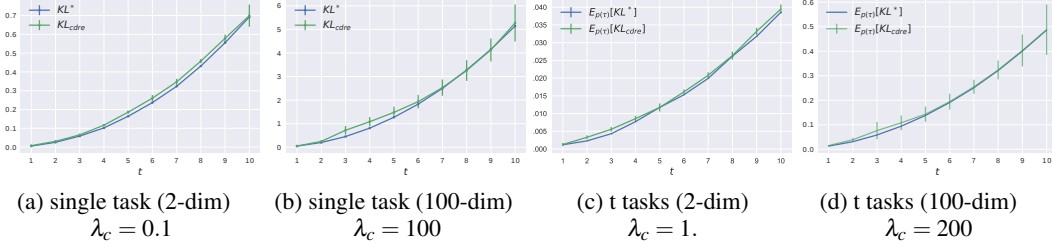

| (a) single task (2-dim) $\lambda_c = 0.1$ | (b) single task (100-dim) $\lambda_c = 100$ | (c) t tasks (2-dim) $\lambda_c = 1.$ | (d) t tasks (100-dim) $\lambda_c = 200$ |

Figure 2: Synthetic experiments of estimating KL-divergence by CDRE (error bars from 5 runs). The $x$-axis is the index of time steps. Figs. 2a and 2b compare the true KL-divergence with estimated KL-divergence for a single task, Figs. 2c and 2d compare the average KL divergence of $t$ tasks at time $t$. $KL^*$ and $KL_{cdre}$ denote the true value and estimated value of KL-divergence, respectively.

In the experiments with synthetic data, we first simulate a single Gaussian distribution drifting over time and estimate the KL-divergence between the model distribution $q_t(x)$ at time $t, \forall t \geq 1$ and the target distribution $p(x)$ seen at time $t = 1$. The distribution $q_t(x) = \mathcal{N}(\mu_t, \sigma_t^2 I)$ where $\mu_t = \mu_0 + \Delta\mu \times t, \sigma_t = \sigma_0 + \Delta\sigma \times t$ and $p(x) = \mathcal{N}(\mu_0, \sigma_0^2 I)$. We set $\mu_0 = 0, \sigma_0 = 1, \Delta\mu = -\Delta\sigma = 0.05$ for 2-dimensional data and $\Delta\mu = -\Delta\sigma = 0.02$ for 100-dimensional data. We set $\Delta\sigma$ to be negative, simulating underestimated variance as it is a common issue in generative models. The sample size of each distribution is 10000. The results are shown in Figs. 2a and 2b. We then simulate the scenario

of continual learning, adding a new target distribution at each time step as learning a new task. In this case, real samples of task $\tau$ are drawn from a Gaussian distribution $p(x|\tau) = \mathcal{N}(\mu_\tau, \sigma_\tau^2 I)$, and the model samples are drawn from $q_t(x|\tau) = \mathcal{N}(\mu_\tau + \Delta\mu \times k, (\sigma_\tau + \Delta\sigma \times k)^2 I)$, where $\mu_\tau = 2\tau, \sigma_\tau = 1, k = t - \tau + 1, \tau \leq t$. Figs. 2c and 2d display the estimated average KL-divergence over all learned tasks in comparison with the true value of averaged KL-divergence. We see that the estimated value are close to the true value of KL-divergence in all cases. The model capacity of the estimator may affect the performance as $t$ increases; however, note that with CDRE we have the flexibility to extend the model architecture since the latest estimator only needs the output of the previous one.

We can also see smaller $\lambda_c$ leading to smaller variance but larger bias for which we have explained in Sec. 3. Besides the effect of the hyperparameter $\lambda_c$, there are two major sources of the variance of the estimated f-divergences: 1). The ratio estimator is a neural network which has no assumption about data distributions and trained by stochastic gradient descent, thus different initializations generated by different random seeds may cause larger variance of the results comparing with FID and KID (FID assumes data distributions are Gaussian distributions and KID fits the first three moments by a polynomial kernel). 2). In the formulation of the ratio estimator, we use finite samples to estimate the expectations, which may bring estimation errors and become another source of variance, especially when the overlapping mass of the two distribution is sparse. This explains why the variance increases while the model distribution getting further to the raw data distribution. In this sense, the variance can also be a criterion of evaluating generative models.

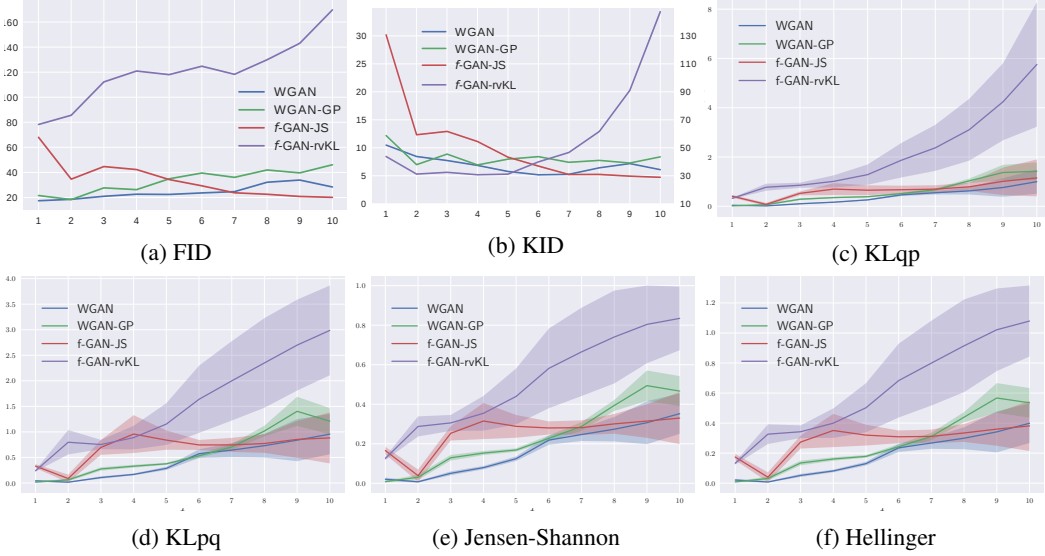

(a) FID  (b) KID  (c) KLqp

(d) KLpq  (e) Jensen-Shannon  (f) Hellinger

Figure 3: Evaluating GANs in continual learning on Fashion-MNIST, features for FID and KID are extracted from the classifier, features for $f$-divergences are generated by the CVAE. The dimension of generated features is 64. The sample size is 6000 for each class. The shaded area are plotted by standard deviation of 10 runs. The y-axis in the right side of Fig. 3b is the y-axis of the purple line ($f$-GAN-rvKL), which is in a much larger scale than others.

## 4.2 EXPERIMENTS WITH IMAGE DATA

Fig. 3 compares several GANs trained on Fashion-MNIST in continual learning using FID, KID and a few members of $f$-divergences. We display randomly chosen model samples of those GANs in Fig. 4. The experiment results of MNIST are shown in the Appx. C.

In general, these measures have a consensus that $f$-GAN-rvKL gives the worst performance during the whole process. They also agree that $f$-GAN-JS is the second worst before task 6 and WGAN-GP is the second worst after task 7. And not surprisingly, there are several disagreements between these measures. Regarding $f$-GAN-JS, FID and KID are decreasing from task 3 to 10 whereas members of $f$-divergences are more like a plateau from task 4 to 10. According to Fig. 4c, we would argue that there is no notable improvement observed from task 3 to 10, which indicates the decreasing trend of FID and KID is doubtful. Moreover, KID of WGAN and WGAN-GP fluctuate around an

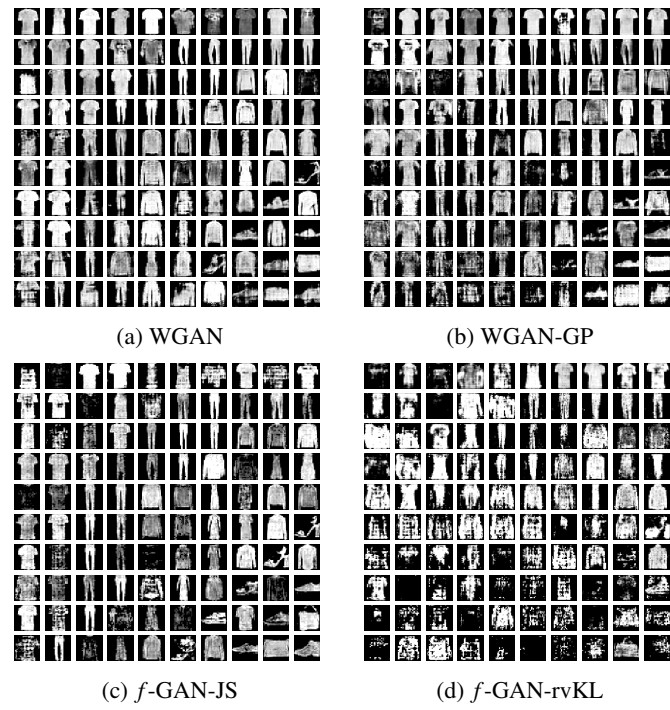

(a) WGAN

(b) WGAN-GP

(c) $f$-GAN-JS

(d) $f$-GAN-rvKL

Figure 4: Fashion-MNIST samples generated by GANs in continual learning. In each figure, each row displays images generated by the model at each task,the order is from the top to bottom (task 1 to 10). The displayed samples are randomly chosen from generated samples of each class.

approximately horizontal line whilst other measures show an increasing trend. Visually, samples from WGAN and WGAN-GP (Figs. 4a and 4b) are obviously losing fidelity while learning more tasks, which matches the increasing trend in all measures except KID. Another disagreement is that WGAN and $f$-GAN-JS are considered as almost equally well from task 8 to 10 by the evaluation from $f$-divergences whereas FID and KID prefer $f$-GAN-JS more than WGAN. In Figs. 4a and 4c, We observe that $f$-GAN-JS generates more images with darker color and lower fidelity than WGAN, however, images with brighter color generated by it show higher fidelity than those samples from WGAN. This ambiguity may result in the disagreement as these measures may focus on different parts of the density mass, which is analogous to the second toy experiment (Fig. 5) in Appx. B. All in all, the experiment results show that $f$-divergences estimated by CDRE do provide meaningful evaluations of generative models in continual learning.

## 5 DISCUSSION

We show that CDRE is capable of estimating $f$-divergences in the setting of continual learning. It provides an alternative approach for model selection when other measures are not possible in continual learning. The results also demonstrate CDRE can work with a simple CVAE for feature generation when a pre-trained classifier is not available. Moreover, our experiments show that CDRE can work well when the differences between model samples and real samples are significant, which is a rather difficult situation for estimating density ratios. We consider a more sophisticated method of dimensionality reduction for CDRE as a future work, making it work more stably with high-dimensional data.

It is also possible to estimate the Bregman divergence by ratio estimation (Uehara et al., 2016), giving even more options of divergences for evaluating generative models. DRE also has many other applications other than estimating $f$-divergences, such as change detection (Liu et al., 2013), mutual information estimation (Sugiyama et al., 2012), etc.. Likewise, CDRE may be useful for more applications in continual learning which we would like to explore in the future.

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

## A  EXPERIMENTAL SETTINGS

We elaborate the experimental settings of our experiments here and all implementations [1] are based on Python 3 and TensorFlow 1.10 (Abadi et al., 2015).

### A.1  CONFIGURATION OF FEATURE GENERATORS

The classifier used to extract features on both datasets has two convolutional layers with filter shape [4,4,1,64], [4,4,64,128] respectively, strides are all [1,2,2,1], and two dense layers with hidden units [6272, 64]. Batch normalization is performed on the second conv layer and first dense layer.

The encoder of the VAE has two dense layers with hidden units [512, 256], output dimension is 64, and decoder has two dense layers with hidden units [256, 256], output dimension is 784. The variance of noise is set to 0.1. It's trained with L2 regularization and the Lagrange multiplier is 0.001.

Both the classifier and VAE are trained with batch size = 100, learning rate = 0.001 and 100 epochs. Activation function of all hidden layers is ReLU. The optimizer is Adam (Kingma & Ba, 2014) for all training runs.

### A.2  CONFIGURATION OF RATIO ESTIMATORS

The ratio estimator we used in all experiments is the log-linear model as defined in Equation 3 and Equation 12, and $\psi(\cdot)$ is a neural network with two dense layers, each having 256 hidden units. It is trained by Adam optimizer and batch size = 2000. On toy data in continual learning, learning rate = 2e-5, $\lambda_c$ are shown in Figure 4, sample size is 10000 per task. On MNIST and Fashion-MNIST in continual learning, $\lambda_c = 1.$, learning rate = 1e-5, sample size of each class is 1000, validation sample size of each class is also 1000. Maximum number of epochs is 1000 for DRE and 2000 for CDRE at each task, the training process could be early stopped when validation loss increasing. $\lambda_c$ increases linearly with the number of tasks in continual learning, which means at task $t$, $\lambda_{c,t} = \lambda_{c,0} \times t$.

### A.3  CONFIGURATION OF GANS

All GANs are trained with a discriminator having the same architecture as the classifier described above, except the last dense layer has 1024 hidden unites; a generator having two convolutional layers with filter shape [4,4,64,128],[4,4,1,64] respectively and two dense layers with hidden units [1024,6272], applied batch normalization on two dense layers. Activation function is leaky ReLu (Xu et al., 2015) for all hidden layers. The optimizers are Adam, batch size is 64 and learning rate is 0.0002 and 0.001 for discriminator and generator respectively. $f$-GAN-rvKL is trained by 6 epochs as longer training makes it worse, all others trained by 15 epochs. The random input of generators has dimension 64 for both MNIST and Fashion-MNIST.

## B  EVALUATING GENERATIVE MODELS USING $f$-DIVERGENCES

We provide experimental results of comparing $f$-divergences with FID (Heusel et al., 2017), KID (Bińkowski et al., 2018) and PRD (Sajjadi et al., 2018) in a few toy experiments and a high-dimensional dataset of the real world. Through these experiments, we show that $f$-divergences can be alternative measures of generative models and one may obtain richer criteria by $f$-divergences.

### B.1  TOY EXPERIMENTS WITH MNIST

We present toy experiments to show the differences between $f$-divergences and FID, KID in terms of evaluating generative models. We demonstrate the experiment results through two most popular members of $f$-divergences: KL-divergence and reverse KL-divergence.

In the first experiment, We have two cases: (i) the target distribution $P$ contains half of the classes of MNIST, and the evaluated distribution $Q$ includes all classes of MNIST; (ii) the reverse of (i).

---

[1]Source code can be found on www.github.com/revealedafterreview

We obtain density ratios by KLIEP and then estimate $f$-divergences by density ratios. The results are shown in Tab. 1, with PRD curves displayed beside the table. Since the KLIEP objective is not symmetric (Eq. (2)), the estimated KL divergences are not symmetric when switching the two sets of samples. As expected, $D_{KL}(P||Q)$ prefers $Q$ with larger recall and vice versa. Neither FID nor KID are able to discriminate between these two scenarios.

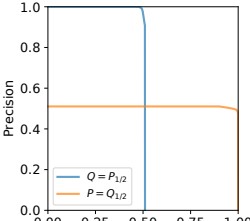

Table 1: Results of the toy experiment. $P = Q_{1/2}$ implies case (i), $Q = P_{1/2}$ implies case (ii). Standard deviations are from 5 runs.

|  | FID | KID | $D_{KL}(P||Q)$ | $D_{KL}(Q||P)$ |
|---|---|---|---|---|
| $P = Q_{1/2}$ | $50.39 \pm 0.00$ | $2.04 \pm 0.01$ | $0.67 \pm 0.00$ | $3.78 \pm 1.22$ |
| $Q = P_{1/2}$ | $50.39 \pm 0.00$ | $2.03 \pm 0.02$ | $2.49 \pm 0.30$ | $2.38 \pm 1.78$ |

In the second experiment, we show that $f$-divergences may provide different opinions with FID and KID in certain circumstances because FID and KID are based on Integral Probability Metrics (IPM) (Sriperumbudur et al., 2012) which focus on parts of the distribution with most mass whereas $f$-divergences are based on density ratios which may give more attention on parts with less mass (due to the ratio of two small values can be very large). To show this, we simulate two sets of model samples by injecting two different types of noise into MNIST data and evaluate them on the original feature space (which is 784 dimensions). Regarding the first type of noise, we randomly choose 50% samples and 1 dimension to be corrupted (set the pixel value to 0.5); for the second one, we

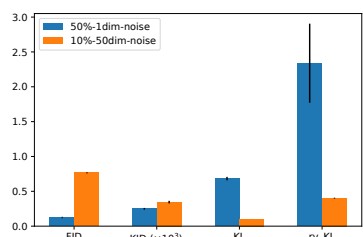

Figure 5: Toy experiments with different types of noise injected into MNIST (error bars from 5 runs).

randomly choose 10% samples and 50 dimensions to be corrupted. The results are shown in Fig. 5, in which KL-divergence and reverse KL-divergence disagree with FID and KID regarding which set of samples is better than the other.

### B.2 EXPERIMENTS WITH HIGH-DIMENSIONAL FEATURES ON FFHQ

Table 2: Evaluating StyleGAN using $f$-divergences estimated by DRE

|  | StyleGAN | Real samples |
|---|---|---|
| KL | $2.47 \pm 0.02$ | $0.02 \pm 9.1e-4$ |
| rv_KL | $3.29 \pm 0.18$ | $0.02 \pm 9.3e-4$ |
| JS | $0.86 \pm 0.01$ | $0.01 \pm 4.5e-4$ |
| Hellinger | $1.04 \pm 0.02$ | $0.01 \pm 4.6e-4$ |

In order to show that DRE can work with high dimensional data with high fidelity, We also conducted an experiment with samples generated by StyleGAN on FFHQ dataset (Karras et al., 2019). We obtain model samples by the pre-trained StyleGAN[2] and estimate $f$-divergences on the inception feature space with 2048 dimensions. The sample size is 50000 and we compare the model samples with real samples. We see that the $f$-divergences estimated by DRE giving reasonable results with small variance (Tab. 2), which indicates it can be an alternative measure for those state-of-the-art generative models.

## C  EXPERIMENT RESULTS ON MNIST

We show experiment results on MNIST in Figs. 6 and 7, the settings of these experiments are the same as experiments with Fashion-MNIST.

---

[2]https://github.com/NVlabs/stylegan

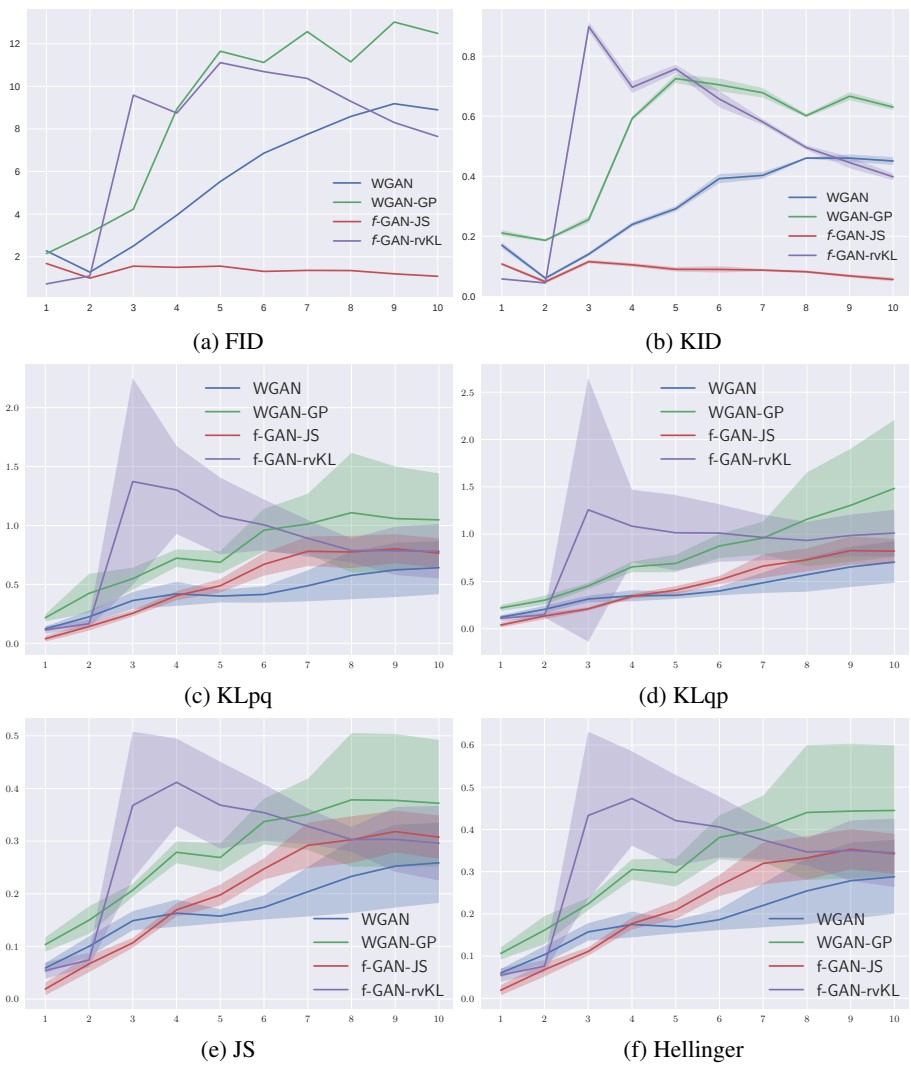

Figure 6: Evaluating GANs in continual learning on MNIST, features for FID and KID are extracted from the classifier, features for $f$-divergences are generated by CVAE. The shaded area are plotted by standard deviation of 10 runs.

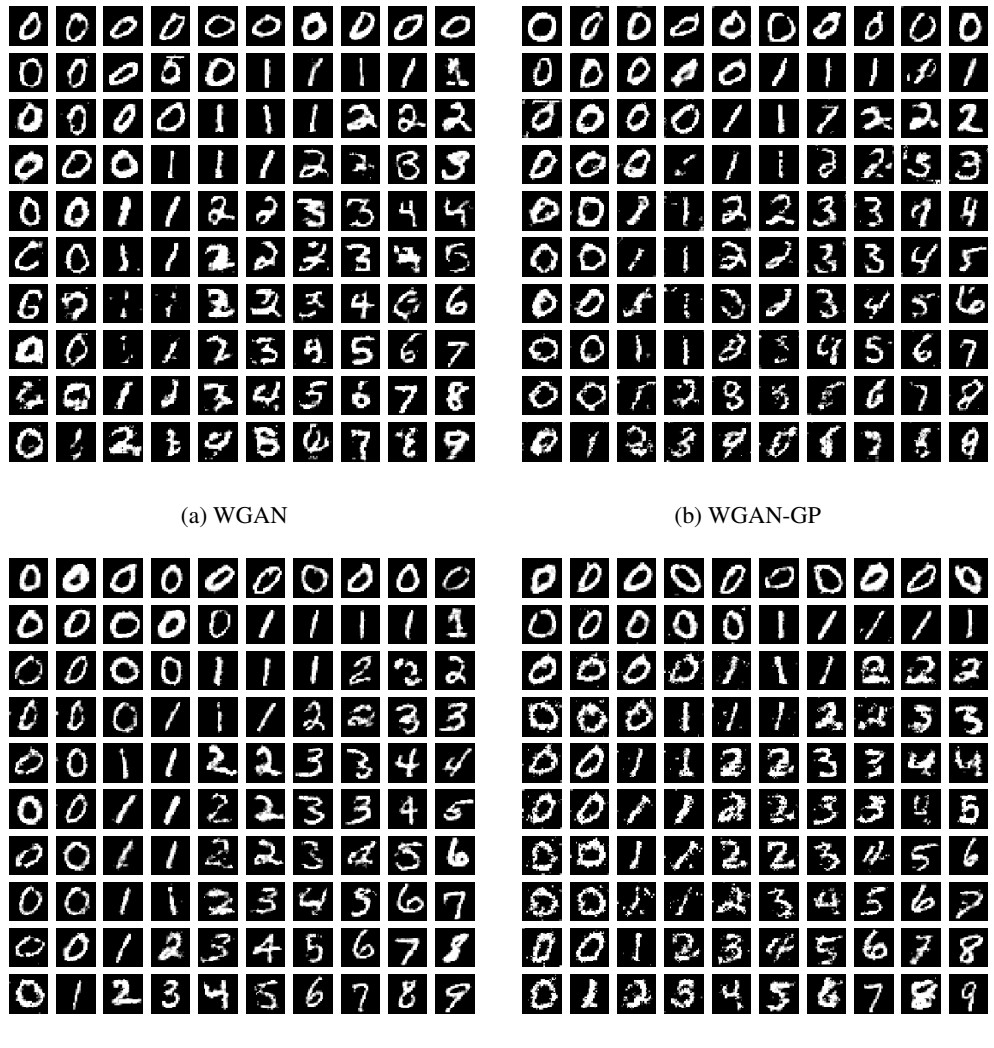

(a) WGAN

(b) WGAN-GP

(c) $f$-GAN-JS

(d) $f$-GAN-rvKL

Figure 7: MNIST samples generated by evaluated GANs in continual learning. In each figure, each row displays figures generated at each task,the order is from the top to bottom.

