# OpenReview forum: "Continual Density Ratio Estimation (CDRE): A new method for evaluating generative models in continual learning"
_ICLR.cc/2020/Conference — Reject_

### Official Review · AnonReviewer1 · 2019-10-23
**Official Blind Review #1**

**Rating:** 1

**Review:**

This submission proposes a continual density ratio estimation method and suggested its application to evaluate a continually learned generative model without requiring the previous training data. The basis of the continual density estimation is based on a recursive relationship between the density ratio at step t and that at t - 1. Continually estimating the density ratio without storing the raw data is an interesting topic and could be useful for continual learning. To my knowledge, I have not seen this in earlier publications.

However, I give reject to this paper because of the following reason:

The writing of this paper is not easy to follow.

- The beginning of section 3 (CDRE in continual learning), I found it difficult to understand why the model q needs to be updated (indexed by t) while p(x) is not dependent on t. As far as I know, under the continual learning setting the data distribution p(x) is also conditioned on t. I interpret it as a general introduction on how density ratio could be estimated continually.
- The Lagrange multiplier and the bias / variance statements need elaboration, I don't understand how it is affecting the bias and variance.
- In the second part of section 3, the continual learning setting is introduced (in equation 11), however, it is no longer reasonable to use the symbol r_t in equation 12 which was initially defined in equation 5.
- A loss for continual VAE is proposed in seciton Feature generation for CDRE, however, the p(x) is again independent of t. And I'm also suspicious that equation 13 is the correct way of adjusting VAE's objective with VCL. In VCL, the KL divergence is on the parameter distribution, which could help prevent forgetting, however, here the KL is between VAE's approximate posteriors, which alone is not sufficient for keeping the information of previous tasks.
- There's lack of analysis / interpretation of results for section 4.1, e.g. what is the motivation of the experiments and what is the conclusion.
- Through out section 4.2 - 4.3, it is not explained what is the source of variance in the experiment results.


**Experience Assessment:**

I have published one or two papers in this area.

**Review Assessment: Checking Correctness Of Derivations And Theory:**

I carefully checked the derivations and theory.

**Review Assessment: Checking Correctness Of Experiments:**

I assessed the sensibility of the experiments.

**Review Assessment: Thoroughness In Paper Reading:**

I read the paper at least twice and used my best judgement in assessing the paper.

---

> ### Author Response · Authors · 2019-11-13
> **Response to R1**
>
> Thanks for reviewing our paper and giving valuable comments. Please find our response to your concerns in the following:
>
> "- The beginning of section 3 (CDRE in continual learning), I found it difficult to understand why the model q needs to be updated (indexed by t) while p(x) is not dependent on t..."
>
> Yes, you are correct, the beginning of Sec. 3 is an introduction to continual density ratio estimation in a general form, where we only consider one target distribution throughout the whole time.  p(x) represents the density function of the target distribution and thus it is independent on t.  In contrast, q_t(x) represents the density function of samples generated by a model at t, which we assume it changes while t changes.  We have added the explanation at the beginning of Sec. 3 in the revision.
>
> "- The Lagrange multiplier and the bias / variance statements need elaboration..."
>
> The density ratio estimator of KLIEP is an asymptotically unbiased estimator when the constraint is satisfied. However, as we replace the hard constraint by a soft constraint, the larger $\lambda_c$ makes the estimator with soft constraint closer to the unbiased one, which leads to smaller bias. The bias is getting less when increasing the lambda, and as a tradeoff, the variance starts to increase. We have added the discussion into the paragraph below Eq. 10 in the revision.
>
> "- In the second part of section 3,..., it is no longer reasonable to use the symbol r_t in equation 12 which was initially defined in equation 5. "
>
> Thanks for pointing out this issue, we should adjust the notations in equation 5 to the setting of continual learning first. We have corrected Eq. 11-14 in the revision.
>
> "- A loss for continual VAE ... here the KL is between VAE's approximate posteriors, which alone is not sufficient for keeping the information of previous tasks. "
>
> Sorry for the confusion of our notations, we have corrected the formulation in Eq.15 in the revision.
> In terms of adjusting the objective of VAE in VCL, we have tried the proposed objective in VCL. However, the encoder of VAE is task-specific in the experiments of VCL, which is computationally costly for a preprocessing component. We tried sharing both encoder and decoder of VAE in VCL across MNIST tasks, it generates very similar latent code z for different digits which cannot preserve the difference between two distributions and causes the estimated f-divergences always small.  We chose the current form of the objective due to its simplicity and effectiveness, nonetheless, it is flexible to deploy other methods for feature generation of CDRE. We have added more discussion in the revision.
>
> "- There's lack of analysis / interpretation of results for section 4.1... "
>
> We compare f-divergences and a few commonly used measures of generative models by a few toy experiments in section 4.1 because there is no prior work discussing evaluating generative models by f-divergences. We show that f-divergences may provide different rankings with FID, KID because it may pay more attention to different parts of density mass, which can be helpful for understanding the experiment results in the later sections. We have moved this part to the appendix in the revision so that the structure of the paper can be clearer.
>
> "- Throughout section 4.2 - 4.3, it is not explained what is the source of variance in the experiment results."
>
> There are two major sources of the variance of the estimated f-divergences:
> 1). The ratio estimator is a neural network which has no assumption about data distributions and trained by stochastic gradient descent, thus different initializations generated by different random seeds may cause larger variance of the results comparing with FID and KID (FID assumes data distributions are Gaussians and KID fits the first three moments by a  polynomial kernel).
> 2). In the formulation of the ratio estimator, we use finite samples to estimate the expectations, which can be another source of variance, especially when the overlapping mass of the two distribution is sparse. This is demonstrated in Fig.3 & 4, the variance increases while the model distribution getting further to the raw data distribution. In this sense, the variance can also be a criterion for evaluating generative models.
> We have added the discussion into Sec. 4 in the revision.

---

### Official Review · AnonReviewer2 · 2019-10-23
**Official Blind Review #2**

**Rating:** 1

**Review:**

######### Updated Review ###########

I'd like to thank the author(s) for their rebuttal. However, I am still on the same boat with R1 and recommend rejection for this submission.


################################


This submission seeks to evaluate generative models in a continual learning setup without storing real samples from the target distribution.  The main technique the author(s) have used is the likelihood-ratio trick. I question the scope of this paper, as this is not a topic of general interest to the community. Additionally, the density ratio estimation technique is fairly standard. I vote to reject this submission for the lack of highlights and relevant potential applications.

My main argument for rejection.
While continual learning is a trendy topic in the AI community, it's less well-received in the context of generative modeling, probably for the lack of real applications. Such works, including this one, fail to address any real challenge, as the hypothesized scenario is unrealistic. For example, I am not convinced of the significance of using f-div to evaluate model performance. And since importance sampling is notorious for its variance issues (the essential mathematical tool used in this model), the estimate is not expected to be reliable, say subsequent tasks q_t and q_{t-1} differ somehow. This submission feels more like playing a game with the rules defined by the author(s), not driven by practical considerations.



**Experience Assessment:**

I have read many papers in this area.

**Review Assessment: Checking Correctness Of Derivations And Theory:**

I assessed the sensibility of the derivations and theory.

**Review Assessment: Checking Correctness Of Experiments:**

I assessed the sensibility of the experiments.

**Review Assessment: Thoroughness In Paper Reading:**

I read the paper at least twice and used my best judgement in assessing the paper.

---

> ### Author Response · Authors · 2019-11-13
> **Response to R2**
>
> First of all, we respectfully disagree with that generative models in continual learning are unrealistic. They certainly have realistic applications.  “Continual learning (CL) is the ability of a model to learn continually from a stream of data, building on what was learned previously, hence exhibiting positive transfer, as well as being able to remember previously seen tasks.” This definition is from the workshop of continual learning in NeurIPS 2018, which has pointed out two advantages of continual learning: 1) enabling positive transferring of knowledge; 2). preventing from forgetting previously learned knowledge. Such abilities are mimicking human being’s learning abilities in the real world and can be beneficial to not only classifiers but also generative models. For example, the numerous applications of GANs can be much more powerful if they succeed in continual learning. Imagining we train a generative model to generate one type pf sounds, it would be more attractive if the model can continuously learn to generate new sounds without retraining on all learned sounds.
>
> Second, generative modeling in continual learning is not less well-received at all. Actually, the most popular works for continual learning ([1,2,3,4]), including methods of parameter regularization and incrementally growing up model architectures, can be applied to generative models as well. For example, in some specific works ([5,6,7]) of GANs in continual learning, authors compare their method with EWC [1] too. Another branch of methods in continual learning is likelihood regularization, among which generative modeling itself is an important direction [8].
>
> Third, we have shown f-divergences can be valid measures of generative models in the experiment results (Sec. 4) by comparing with other commonly used measures. Please also refer to the response to the last question from R1 for the variance issues.
>
> [1]. Kirkpatrick, James, et al. "Overcoming catastrophic forgetting in neural networks." Proceedings of the national academy of sciences 114.13 (2017): 3521-3526.
> [2]. Zenke, Friedemann, Ben Poole, and Surya Ganguli. "Continual learning through synaptic intelligence." Proceedings of the 34th International Conference on Machine Learning-Volume 70. JMLR. org, 2017.
> [3]. Cuong V Nguyen, Yingzhen Li, Thang D Bui, and Richard E Turner. Variational continual learning.  In International Conference on Learning Representations, 2018.
> [4]. Schwarz, Jonathan, et al. "Progress & Compress: A scalable framework for continual learning." International Conference on Machine Learning. 2018.
> [5]. Wu, Chenshen, et al. "Memory replay GANs: Learning to generate new categories without forgetting." Advances In Neural Information Processing Systems. 2018.
> [6]. Ostapenko, Oleksiy, et al. "Learning to Remember: A Synaptic Plasticity Driven Framework for Continual Learning." Proceedings of the IEEE Conference on Computer Vision and Pattern Recognition. 2019.
> [7]. Lesort, Timothée, et al. "Generative models from the perspective of continual learning." 2019 International Joint Conference on Neural Networks (IJCNN). IEEE, 2019.
> [8]. Shin, Hanul, et al. "Continual learning with deep generative replay." Advances in Neural Information Processing Systems. 2017.

---

### Official Review · AnonReviewer3 · 2019-10-23
**Official Blind Review #3**

**Rating:** 6

**Review:**

In this paper, authors propose a continual learning for density-ratio estimation. The formulation of the CDRE Eq.(5)is quite intuitive, and it makes sense. Then, a log-linear model is employed for a density-ratio model, and it is estimated by using a density-ratio estimation algorithm. Through experiments, the authors show that the proposed algorithm outperforms existing methods.

The paper is clearly written and easy to read. The density-ratio estimation algorithm for continual learning is new and interesting.

Detailed comments:
1. I am pretty new to the continual learning. The formulation of CDRE is interesting. However, I am still not that certain whether the setup is realistic. In the introduction, authors describe that we cannot obtain data points due to privacy or limited cost budget. More specifically, if it is a privacy issue, we may not be able to use the model trained by the private data as well. Also, could you give me a couple of examples of the limited cost budget case?

2. In this paper, authors employed the log-linear model. If we use another model, performance can be changed?



**Experience Assessment:**

I have published one or two papers in this area.

**Review Assessment: Checking Correctness Of Derivations And Theory:**

I assessed the sensibility of the derivations and theory.

**Review Assessment: Checking Correctness Of Experiments:**

I assessed the sensibility of the experiments.

**Review Assessment: Thoroughness In Paper Reading:**

I read the paper at least twice and used my best judgement in assessing the paper.

---

> ### Author Response · Authors · 2019-11-13
> **Response to R3**
>
> Thanks for reviewing our paper and giving valuable comments. Please find our response to your concerns in the following:
>
> "1. ... In the introduction, authors describe that we cannot obtain data points due to privacy or limited cost budget. More specifically, if it is a privacy issue, we may not be able to use the model trained by the private data as well. Also, could you give me a couple of examples of the limited cost budget case? "
>
> There can be diverse reasons in the real world of limiting access to the raw data after a model is trained on it. For example, researchers of a hospital may have trained a model for one type of disease and the raw data of patients cannot be shared with other institutions, if they want to collaborate with another institution to enable the model capable of a similar type of disease as well, the model can only be incrementally trained on new data without sharing the previous data. Sharing the model is a lot less sensitive than directly sharing data points, though the trained model still contains information of the original data. If this is still not allowed, we may be able to use some techniques such as importance sampling to estimate the results on the original data set by another publishable data set (i.e. E_{q(x)}[p(x)/q(x) L(x)], where L(x) is the loss function of the model).  Besides the privacy issue, a limited cost budget can be another cause of such a problem, such as the data storage cost is quite high, or the data is not available for free after its copyright has expired. We have added the examples in the introduction section of the revision.
>
> "2. In this paper, authors employed the log-linear model. If we use another model, performance can be changed?"
>
> Yes, if using other types of models in the ratio estimator, the performance can be different.  We also tried ratio estimators defined as in f-GAN [1] and found they are less robust when the difference of two distributions is significant. The performance can also be affected by different usage of the estimated ratios. As we apply it to estimate f-divergences and many of the popular members of f-divergences based on log-ratios, the log-linear model is a suitable choice. On the other hand, if estimating Pearson \chi^2 divergence which based on square loss of ratios, the linear model suggested in [2] may be better. We have put some discussion in Sec.3 after Eq. 10 in the revision.
>
> [1]. Nowozin, Sebastian, Botond Cseke, and Ryota Tomioka. "f-GAN: Training generative neural samplers using variational divergence minimization." Advances in neural information processing systems. 2016.
> [2]. Kanamori, Takafumi, Shohei Hido, and Masashi Sugiyama. "A least-squares approach to direct importance estimation." Journal of Machine Learning Research 10.Jul (2009): 1391-1445.

---

> > ### Comment · AnonReviewer3 · 2019-11-15
> > **Thank you for the response**
> >
> > Dear authors
> >
> > We thank the author for the response.
> >
> > >There can be diverse reasons in the real world of limiting access to the raw data after a model is trained on it. For example, researchers of a hospital may have trained a model for one type of disease and the raw data of patients cannot be shared with other institutions, if they want to collaborate with another institution to enable the model capable of a similar type of disease as well, the model can only be incrementally trained on new data without sharing the previous data. Sharing the model is a lot less sensitive than directly sharing data points, though the trained model still contains information of the original data. If this is still not allowed, we may be able to use some techniques such as importance sampling to estimate the results on the original data set by another publishable data set (i.e. E_{q(x)}[p(x)/q(x) L(x)], where L(x) is the loss function of the model).  Besides the privacy issue, a limited cost budget can be another cause of such a problem, such as the data storage cost is quite high, or the data is not available for free after its copyright has expired. We have added the examples in the introduction section of the revision.
> >
> > I am still not sure whether this is truly a realistic case. Basically, if we really need to solve healthcare problems, the best way would be to build a database and use supervised algorithm (of course, it may be difficult). Otherwise, the machine learning method is basically useless (as reviewer #2 pointed out, this sounds a problem for machine learning method). It would be great that authors can demonstrate this in the paper. Then, the paper is very interesting and important.
> >
> > Overall, I still like the density-ratio formulation itself. However, I am not sure the application of the approach is useful at this point.

---

> > > ### Author Response · Authors · 2019-11-15
> > > **Thanks for replying to our response**
> > >
> > >
> > > Dear Reviewer,
> > >
> > > We really appreciate your feedback on our response. We understand your concern, but we also would like to point out that generative models are not only including unsupervised approaches (like GANs or VAEs) but also including many probabilistic latent variable models that can be supervised approaches whilst we can draw samples from it. Regarding the scope of health care, it's usually difficult to obtain a large dataset for a specific problem and requires some explanation of the results, in such cases probabilistic modeling is often a better choice than deep learning methods. We show experiments on GANs and image data because they are widely used bench-mark tasks in continual learning, however, our method can be applied to any generative models as long as we can draw samples from the model.

---

### Decision · Program_Chairs · 2019-12-19

**Decision:**

Reject

**Comment:**

The paper seems technically correct and has some novelty, but the relevance of the paper is questionable. Considering the selectiveness of ICLR, I cannot recommend the paper for acceptance at this point.

In more detail: the authors propose a technique for estimating density rations between a target distribution of real samples and a distribution of samples generated by the model, without storing samples. The method seems to be technically well executed and verified. However, there was major concerns among multiple reviewers that the addressed problem does not seem relevant to the ICLR community. The question addressed seemed artificial, and it was not considered realistic (by R2 and also by R1 in the confidential discussion). R3 also expressed doubts at the usefulness of the method.

Furthermore, some doubts were expressed regarding clarity (although opinions were mixed on that) and on the justification of the modification of the VAE objective to the continual setting.